# Human interictal epileptiform discharges are bidirectional traveling waves echoing ictal discharges

Elliot H Smith[1,2]*, Jyun-you Liou[3], Edward M Merricks[2], Tyler Davis[1], Kyle Thomson[4], Bradley Greger[5], Paul House[6], Ronald G Emerson[7], Robert Goodman[8], Guy M McKhann[9], Sameer Sheth[10], Catherine Schevon[2], John D Rolston[1]*

[1]Departments of Neurosurgery and Biomedical Engineering, University of Utah, Salt Lake City, United States; [2]Department of Neurology, Columbia University, New York, United States; [3]Department of Anesthesiology, Weill Cornell Medicine, New York CIty, United States; [4]Department of Pharmacology & Toxicology, University of Utah, Salt Lake City, United States; [5]Department of Bioengineering, Arizona State University, Tempe, United States; [6]Neurosurgical Associates, LLC, Murray, United States; [7]Hospital for Special Surgery, New York, United States; [8]Lenox Hill Hospital, New York, United States; [9]Department of Neurological Surgery, Columbia University Medical Center, New York, United States; [10]Department of Neurological Surgery, Baylor College of Medicine, Houston, United States

*For correspondence:
e.h.smith@utah.edu (EHS);
john.rolston@utah.edu (JDR)

**Abstract** Interictal epileptiform discharges (IEDs), also known as interictal spikes, are large intermittent electrophysiological events observed between seizures in patients with epilepsy. Although they occur far more often than seizures, IEDs are less studied, and their relationship to seizures remains unclear. To better understand this relationship, we examined multi-day recordings of microelectrode arrays implanted in human epilepsy patients, allowing us to precisely observe the spatiotemporal propagation of IEDs, spontaneous seizures, and how they relate. These recordings showed that the majority of IEDs are traveling waves, traversing the same path as ictal discharges during seizures, and with a fixed direction relative to seizure propagation. Moreover, the majority of IEDs, like ictal discharges, were bidirectional, with one predominant and a second, less frequent antipodal direction. These results reveal a fundamental spatiotemporal similarity between IEDs and ictal discharges. These results also imply that most IEDs arise in brain tissue outside the site of seizure onset and propagate toward it, indicating that the propagation of IEDs provides useful information for localizing the seizure focus.

## Editor's evaluation

This manuscript describes the propagation patterns of electrical activity in the brains of patients with drug-resistant epilepsy. Specifically, the authors demonstrate that interictal spikes, commonly observed electrical events in epileptic patients, propagate in a similar manner to seizures, which are relatively uncommon and more difficult to capture. This suggests that interictal spikes could be used in surgical planning, improving the localization and treatment of epileptic networks.

## Introduction

While seizures are mostly unpredictable and rare, electrical recordings from people with epilepsy often show isolated epileptiform discharges between seizures (*Alarcon et al., 1997*; *Jefferys and Avoli, 2012*; *Tatum et al., 2016*). These IEDs are far more frequent, occurring up to several times per minute, and exhibit multidien variation in their frequency that correlates with seizure likelihood, making IEDs an attractive personalized biomarker for seizure risk (*Baud et al., 2018*). Beyond such temporal information about seizure occurrence, there is some evidence for overlap between cortical areas where seizures originate and those with more IEDs (*Alarcon et al., 1997*; *Conrad et al., 2020*; *Marsh et al., 2010*). Furthermore, some retrospective studies showed that removing brain areas with more IEDs improved surgical outcomes in patients with medically refractory epilepsy (*Kim et al., 2010*; *Smart et al., 2012*). Despite these findings, the long-debated relationship between IEDs and seizure generating tissue remains unresolved (*Jefferys and Avoli, 2012*; *Paolicchi et al., 2000*; *Tonini et al., 2004*; *Vakharia et al., 2018*).

Microelectrode array recordings in epilepsy patients have revealed the spatiotemporal features of ictal self-organization (*Eissa et al., 2017*; *Martinet et al., 2017*; *Smith et al., 2020*; *Schevon et al., 2012*; *Smith et al., 2016*). These studies reported two classes of recordings, one in which neuronal firing is recruited into the ongoing seizure, and another in which neuronal firing is relatively unaffected, despite seizure-like field potentials appearing on the same microelectrodes. These classes correspond to two dynamically evolving regions known as the ictal *core* and *penumbra*, respectively (*Schevon et al., 2012*). A slowly-propagating, narrow band of tonic action potential firing, the ictal wavefront (IW), delineates the transition between the core and penumbra (*Martinet et al., 2015*; *Schevon et al., 2012*; *Trevelyan et al., 2006*; *Trevelyan et al., 2007*). These dynamic seizure regions exhibit distinct spatial features. The slowly traveling IW repetitively emits rapidly traveling ictal discharges backwards, toward the seizure core (*Figure 1—figure supplement 1*, *Figure 1—video 1*). These discharges occur following the passage of the ictal wavefront and have thus been termed 'post-recruitment' discharges (*Smith et al., 2016*). In some patients, earlier ictal discharges, termed 'pre-recruitment', are also emitted outward, toward the penumbra (*Martinet et al., 2017*; *Smith et al., 2020*; *Smith et al., 2016*). To avoid confusion between ictal discharges and interictal discharges, we will from here on refer to ictal discharges as seizure discharges (SDs).

The discovery of these spatiotemporal features of human seizure activity inspired a computational model designed to explain the neuronal underpinnings of seizure dynamics from biophysical principles (*Liou et al., 2020*). After several induced seizures in this model, the network produces spontaneous seizures and IEDs with spatiotemporal dynamics. One prediction of the model is that the repeated barrages of traveling synaptic activity during SDs eventually coopt mechanisms of synaptic plasticity, biasing local tissue to propagate IEDs in similar directions as SDs, that is in the opposite direction of the slow propagation of seizure expansion. In this study we test the resultant hypothesis that IEDs have a predominant direction of propagation, towards the site of seizure onset; opposite the direction of seizure expansion. We found that the data supported this hypothesis, and further, that in the majority of participants, IEDs traveled bimodally on a linear axis with predominant and auxiliary sub-distributions whose directions, speeds, and proportions echoing those of SDs. Finally, we directly quantified the extent to which IED directions could be used to predict SD directions.

## Results

### IED detection in human microelectrode array recordings

To examine the spatiotemporal propagation of IEDs, we used a multi-institutional dataset of Utah-style microelectrode array (UEA; 10 × 10 microelectrodes in 4 × 4 mm grid, penetrating 1 mm) recordings from 10 epilepsy patients (two female, μ ±σ age: 29 ± 5.24 years) undergoing monitoring for neurosurgical treatment of medically refractory epilepsy (clinical details in *Appendix 1—table 1*). In order to capture seizures (2.2 ± 1.6 seizures recorded per participant; 22 total), we recorded data continuously throughout the patients' monitoring periods (*Figure 1A*; 4.3 ± 2.4 days per participant; 43 total). Searching through weeks' worth of microelectrode data, we detected 45,623 candidate IEDs across the 10 participants (4562.3 ± 5171.7 per participant) using an IED detection algorithm designed for microelectrode recordings, that operated on features of IEDs based on the American Clinical Neurophysiological Society's definition, namely high-amplitude bursts of beta-range (20–40 Hz) local field

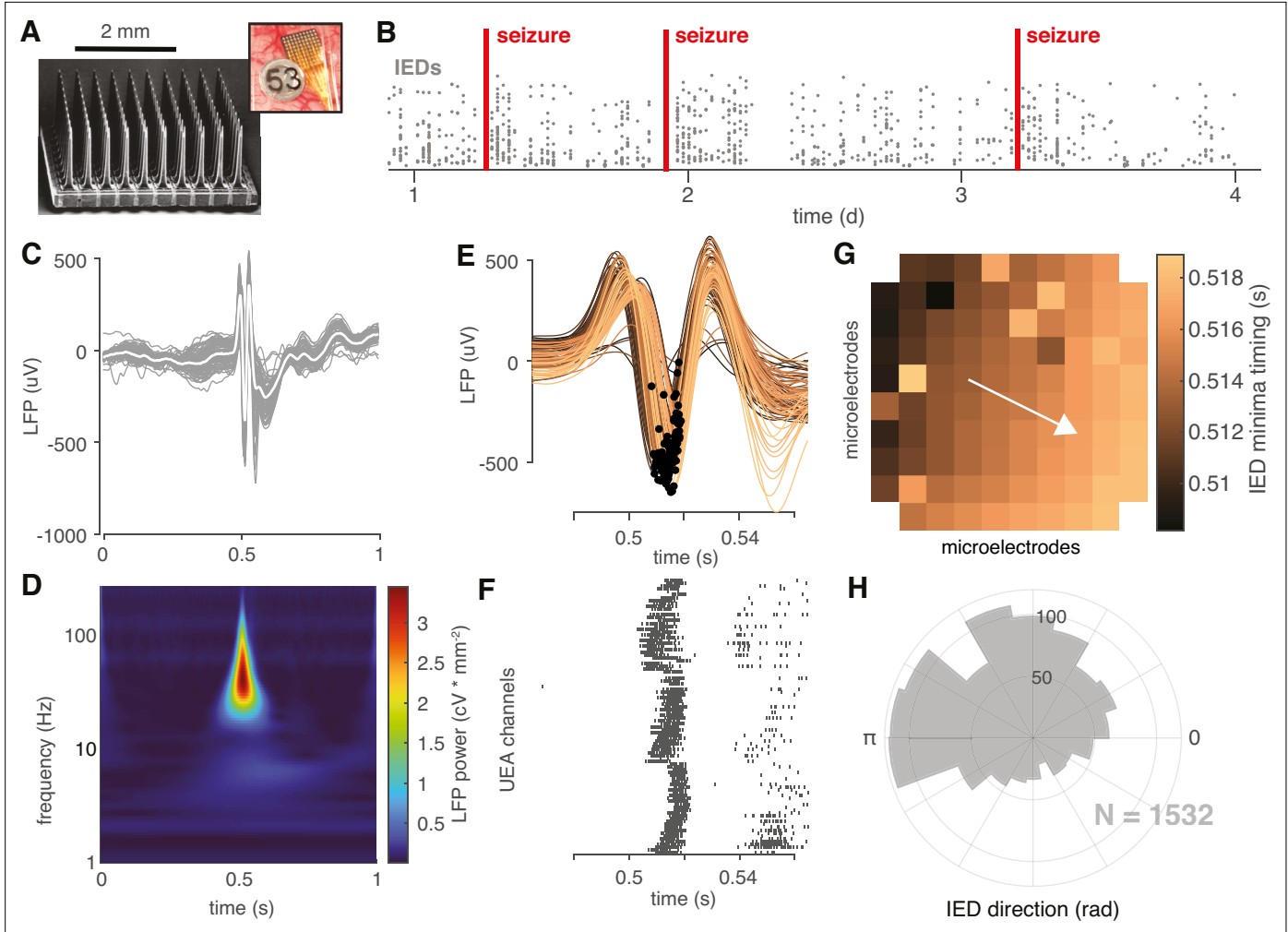

**Figure 1.** IEDs are traveling waves. (**A**) A electron micrograph of an UEA and a picture of an UEA implanted next to an ECoG electrode. (**B**) A raster plot showing an example time course of semi-chronic microelectrode recording during an epilepsy patient's hospital stay. Each gray dot represents the time of one IED (y-axis is arbitrary). (**C**) An example IED recorded across microelectrodes. Each gray line is the same IED recorded on a different microelectrode. The mean IED waveform is overlaid in white. (**D**) Mean spectrogram of the IED shown in (**C**) across microelectrodes. (**E**) A temporally expanded view of the IED shown in (**C**) color coded by when the IED occurs. Black dots indicate the location of the IED negative peaks for each microelectrode. (**F**) A raster plot of IED-associated MUA firing for the same IED as in (**C**) and the same timescale shown in (**F**). (**G**) IED voltage minima timings, color-coded as in (**E**), superimposed across the footprint of the UEA. A white velocity vector derived from the multilinear regression model is also shown on the UEA footprint. (**H**) A polar histogram showing the distribution of all IED traveling waves from which the IED in (**C**) was taken. See *Figure 2—figure supplement 1* and Appendix 1-Algorithm 1 for IED detection and traveling wave classification details.

The online version of this article includes the following video and figure supplement(s) for figure 1:

**Figure supplement 1.** Schematic of seizure domains and hypothesized interictal dynamics.

**Figure supplement 2.** IED detection and artifact rejection.

**Figure supplement 3.** Classifying IED traveling waves.

**Figure supplement 4.** Non-traveling IEDs.

**Figure 1—video 1.** Video of IEDs and ictal recruitment.

https://elifesciences.org/articles/73541/figures#fig1video1

potential (LFP) power occurring across multiple microelectrodes (*Figure 1—figure supplement 2*; Appendix1 - algorithm 1) (*Tatum et al., 2016*). We evaluated the positive predictive value of the algorithm against the ratings of two clinician experts yielding a μ ±σ precision of 0.86 ± 0.04. Inter-rater reliability (Cohen's Kappa) was 44.9%, which is similar to that reported across a large multicenter study of IED ratings (*Jing et al., 2020*). Interrater reliability between the algorithm and board-certified

**Table 1.** Summary statistics for spatiotemporal features of the dataset.

| Participant | Seizure class | N detected IEDs | N (%) traveling waves (LFP) | N traveling waves (MUA) | Median speed (cm/s) | Bimodal? | N seizures |
|---|---|---|---|---|---|---|---|
| 1 | recruited | 1,761 | 1,567 (89.0) | 640 | 20.9 | yes | 1 |
| 2 | recruited | 1,532 | 1,217 (79.4) | 131 | 59.2 | no | 3 |
| 3 | recruited | 17,988 | 10,220 (56.8) | 10,380 | 25.3 | yes | 1 |
| 4 | recruited | 2,806 | 1,429 (50.9) | 284 | 63.7 | yes | 1 |
| 5 | recruited | 2,148 | 1,538 (71.6) | 1,131 | 108.3 | yes | 2 |
| 6 | recruited | 3,502 | 3,143 (89.7) | 2006 | 69 | yes | 2 |
| 7 | penumbral | 4,348 | 2,132 (49.0) | 2,236 | 76.8 | yes | 5 |
| 8 | penumbral | 8,369 | 7,351 (87.8) | 4,977 | 134.7 | no | 0 |
| 9 | penumbral | 834 | 296 (35.5) | 50 | 80.5 | yes | 3 |
| 10 | penumbral | 2,335 | 1,385 (59.3) | 322 | 70.7 | yes | 4 |
| Totals | | 45,623 | 30,278 | 22,157 | | | 22 |

neurologist was 61.1%. Using this algorithm, we detected an average of 0.43 ± 0.51 IEDs per minute. The UEA enabled us to record both LFP data and multiunit action potential firing (MUA) across high-density spatial grid during each IED. These features of an example IED are shown in *Figure 1C–G*.

## IEDs propagate in predominant and auxiliary directions

In order to determine whether the detected IEDs were traveling waves, and to measure wave speeds and directions, we fit a plane to the timings of both IED voltage extrema and MUA event times measured on each microelectrode using multi-linear regression (*Liou et al., 2017*). IEDs with regression slopes that were significantly different from zero were classified as traveling waves (permutation test against a distribution of 1000 spatially permuted timings; *Figure 1H*, *Figure 1—figure supplement 3*). Traveling wave speeds and directions were then derived from each significant model's slope. Based on this operational definition, 30,278 IEDs (3027.8 ± 3190.0 per participant) were classified as traveling waves (66.4%). Example non-traveling IEDs and regression model betas are shown in *Figure 1—figure supplement 4*.

Summary statistics for the spatiotemporal features of IEDs are shown in *Table 1*. Mean IED speeds were on the same order as SDs before the passage of the ictal wavefront (*Liou et al., 2017*; *Smith et al., 2016*). Traveling waves were also detected from MUA, independent of LFP recordings, though at a slightly reduced rate (2215.7 ± 3237.6 per participant; 22,157 total; 48.6%; $\chi^2 = 2957$, p < 0.05). This result was expected, as LFP is a more reliable signal to record, and action potential firing during IEDs has previously been shown to be remarkably heterogeneous, particularly in areas further from the seizure onset zone (*Keller et al., 2010*). That there were significantly more IED traveling waves in UEA recordings that were eventually recruited into the seizure core, further supports the idea that more firing, closer to the seizure onset zone improves reliability of traveling wave detection with MUA (McNemar Test, $\chi^2(1) = 2957$, p < $10^{-6}$). We therefore focus our analysis on IED traveling waves measured from LFP minima in order to understand IED propagation across participants.

Having determined the majority of IEDs met the criteria to be classified as traveling waves, we next sought to understand whether IEDs from each participant exhibited a predominant propagation direction. We therefore tested whether distributions of IED traveling wave directions deviated from a uniform circular distribution (*Fisher, 1953*), we found that each participant's IED traveling wave distribution exhibited a dominant direction (*Figure 2A*; Hermans-Rasson Tests, 1000 permutations, all p < $10^{-3}$). These results show that many IEDs are traveling waves with predominant, consistent directions of travel in each participant.

In addition to a predominant direction common to all participants, many participants appeared to have a second, auxiliary, distribution of IED directions. We therefore fit each participant's IED distribution into a mixture of two circular normal sub-distributions (von Mises distribution). The mixture model

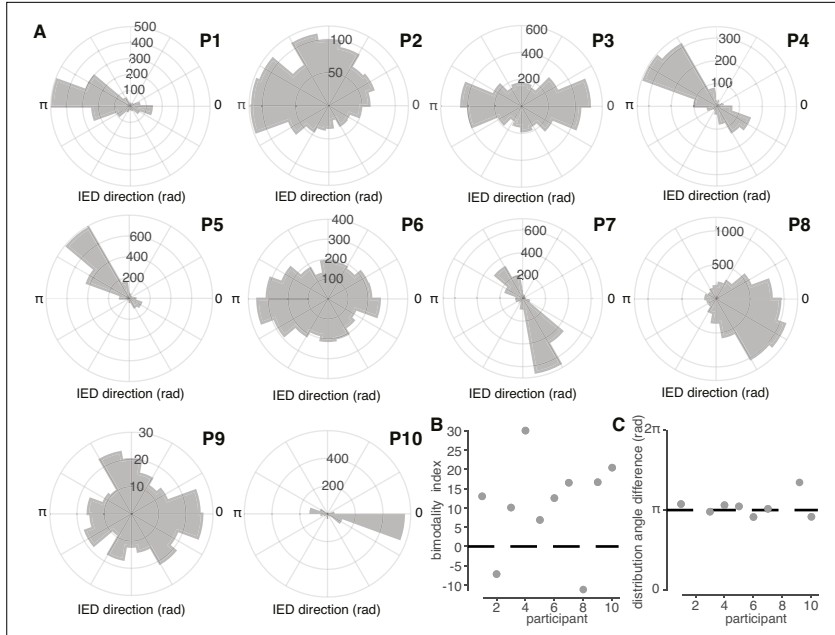

**Figure 2.** IED traveling wave distributions are non-uniform and bimodal. (**A**) Polar histograms of IED traveling wave directions for all 10 participants. Each participant number is indicated in bold above and to the right of each histogram. (**B**) Classification index for bimodality of IED distributions across subjects. Criterion is indicated with a dashed line. (**C**) Difference in median angles of sub-distributions for bimodal IED distributions, with non-bimodal subjects omitted. See *Figure 3* for bimodality classification details.

The online version of this article includes the following figure supplement(s) for figure 2:

**Figure supplement 1.** Procedures for clustering and evaluating the goodness-of-fit of overall and von Mises Mixture distributions.

was compared to a single von Mises distribution model by using permutation-based Kuiper tests (*Figure 2—figure supplement 1*; see Materials and methods). IED traveling wave distributions were thus classified as bimodal in 8 of the 10 participants (*Figure 2B*). The mean and ±s.d. angles between the two IED sub-distributions was 177.9 and 10.7 degrees, respectively (*Figure 2C*). These results show that IEDs also frequently propagate antipodally to their predominant direction, suggesting that IEDs may travel both directions on a linear track through a fixed recording site.

## Spatial features of IED distributions echo ictal self-organization

We next sought to understand whether IED speed and direction related to the spatial self-organization of seizures. We hypothesize that spatial features of IED traveling waves would correlate with seizure propagation direction and those of seizure discharges. We therefore measured the spatial features of seizures first. Fast and slow spatial features of focal seizures were measured in both ictal LFP and MUA bands as in previous reports (*Liou et al., 2017*; *Schevon et al., 2012*; *Smith et al., 2016*). Both of these features were measured using the same multilinear regression framework used to measure IED speed and direction.

Following previous reports with microelectrode arrays, we confirmed that seizures could be divided into two classes based on ictal recruitment: 'recruited' and 'penumbral' (see Materials and methods; *Khodagholy et al., 2015*; *Martinet et al., 2017*; *Merricks et al., 2021*). Recruited tissue exhibited a slow expansion of tonic neuronal firing, the ictal wavefront (*Figure 3D–E*; *Figure 3—figure supplement 1A-D*), followed by rapidly traveling SDs. Penumbral tissue showed neither an IW nor repetitive SDs associated with phase-locked firing (*Figure 3—figure supplement 1E-H*). Six participants' microelectrode arrays were recruited into the ictal core (10 seizures; *Figure 3A–C*), while the four remaining participants were penumbral (12 seizures).

As predicted by our theoretical work (*Liou et al., 2020*), similar patterns of IED and SD propagation were apparent in the majority of seizures in participants with 'recruited' seizures. The majority of IEDs

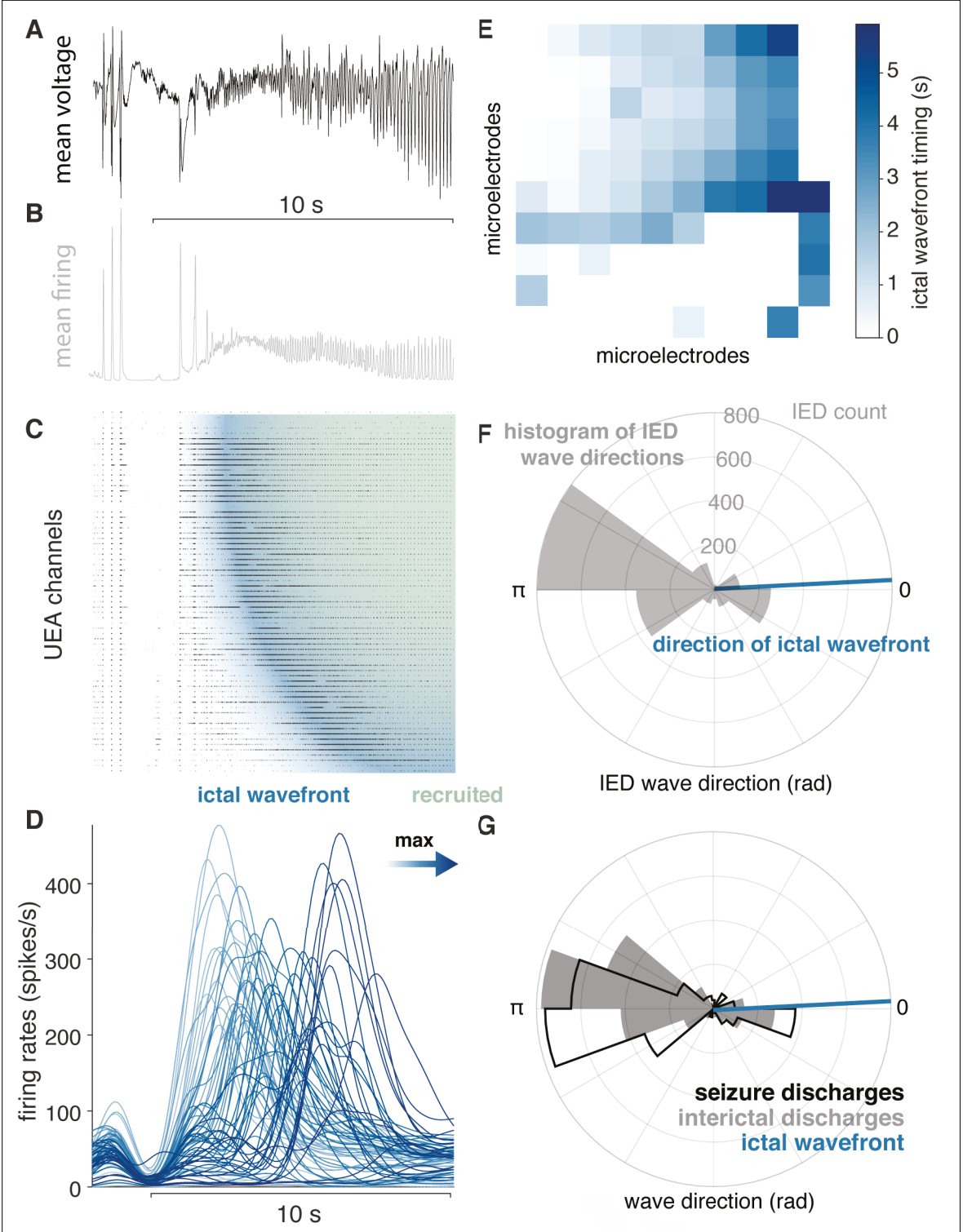

**Figure 3.** IEDs reflect ictal self-organization. (**A**) Mean voltage recorded across the UEA at the start of a seizure. (**B**) Mean MUA firing rate across microelectrodes. (**C**) A raster plot ordered by time of recruitment and color coded to show the IW (blue) and ictal core ("recruited", pink). (**D**) Slow firing rate dynamics on each microelectrode colored by time of maximum firing rate. (**E**) Times of maximum firing rate on each microelectrode superimposed on the footprint of the UEA, color-coded as in (**D**). (**F**) A polar histogram of IEDs and the direction of the IW. (**G**) Polar histograms showing probability densities of IEDs and SDs, and the direction of the IW. See **Figure 4** for examples of classes of microelectrode array recorded ictal self-organization.

The online version of this article includes the following figure supplement(s) for figure 3:

**Figure supplement 1.** Examples of each class of microelectrode seizure recording.

travelled opposite the direction of seizure expansion (i.e. the IW; direction difference from IEDs = 148.9 ± 17.2 degrees; median tests between IW and IED distributions, all p < 0.05; *Figure 3F*). Moreover, IEDs traveled in similar directions as SDs in these participants (example in *Figure 3G*; mean ± s.d. angle difference across participants = 23.7 ± 33.7 degrees). Direction distributions for IEDs, SDs, relative to the direction of the ictal wavefront are shown for all 'recruited' participants in *Figure 4A–F*, and direction summaries for these participants are shown in *Figure 4G–H* (raw directions shown in *Figure 4—figure supplement 1*). In the 'penumbral' category, where the tissue under the UEAs were not obviously recruited from adjacent cortex as in the 'recruited' category, we could not reliably detect or measure the direction of seizure expansion.

In order to directly quantify information about the full distribution of SDs that is gained from observing IEDs, we measured the Kullback-Leibler Divergence (KLD) between IED and SD distributions for each participant. The μ ± σ KLD across these 10 seizures was 0.66 ± 0.56, and are shown next to each pair of distributions in orange in *Figure 4*. The KLD values between IED and SD distributions were all significantly greater than would be expected to occur by chance in each patient (permutation tests, all p < 0.01). These results indicate that less than one extra bit of information is needed to encode the direction of SDs with IEDs on average, suggesting that IED directions could be used to accurately predict SD directions.

## Spatial features of IED sub-distributions predict those of SD sub-distributions

Finally, we sought to further understand geometric features of bimodal IED distributions and how they related to patterns of SD propagation. Such an understanding was only relevant for the 'recruited' participants with bimodal IED distributions (five participants, seven seizures). Using the same bimodality classification strategy as for IEDs, we found that five of the seven these seizures in these participants exhibited bimodal SD distributions, with similarly antipodal sub-distribution directions (*Figure 5A–B*; μ ± σ angle difference = 135.2 ± 24.6). Only participants with bimodal IED distributions had bimodal SD distributions, and only one participant, who had the least bimodal IED distribution, did not have clearly bimodal SDs.

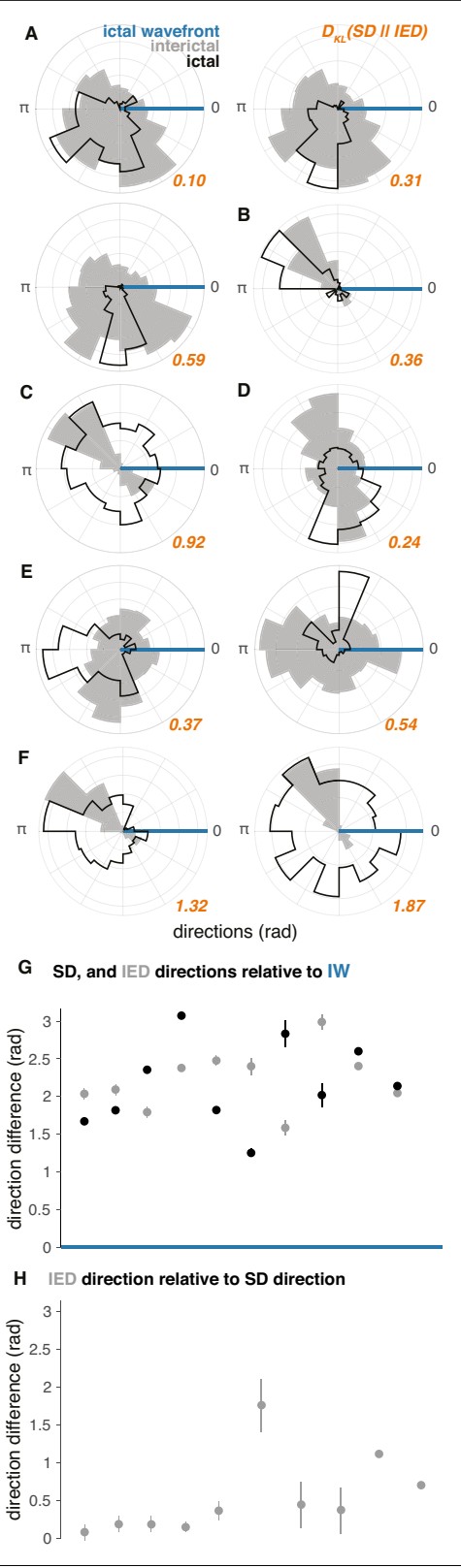

**Figure 4.** IED and SD distributions in 'recruited' UEA recordings. (**A–F**) Each lettered subpanel corresponds to one participant. Each polar histogram corresponds to one seizure. IED distributions are shown in gray and

*Figure 4 continued on next page*

*Figure 4 continued*

SD distributions are shown in black. Both IED and SD distributions are plotted relative to the direction of the IW (blue line). See *Figure 5* for raw IED, SD, and IW directions. (**G**) Direction difference summaries for each seizure ordered and color coded as in (**A–F**). Dots indicate median directions and lines indicate standard deviations. (**H**) Median and standard deviation IED direction summaries relative to median SD directions. See *Figure 5* for raw IW, IED, and SD directions.

The online version of this article includes the following figure supplement(s) for figure 4:

**Figure supplement 1.** Raw IED and SD distributions in 'recruited' UEA recordings.

**Figure supplement 2.** Qualitative relationship between IED propagation and epileptogenic zone in ECoG.

In order to determine whether IEDs in each sub-distribution came from neurophysiologically distinct IED populations, we tested for differences between IED waveforms and firing rates between IED sub-distributions. Such differences might indicate that each IED sub-distribution reflected a separate population of IEDs propagating across the footprint of the UEA. However, neither firing rates nor IED waveforms differed between IED sub-distributions (*Figure 5C–D*; cluster-based permutation tests, all p > 0.05), indicating that neither IED waveforms nor firing rates between the two sub-distributions could be statistically distinguished.

Differences between speeds and proportions of IED sub-distributions, corresponding to those we previously showed in SD sub-distributions (*Smith et al., 2016*), would support a learned relationship between IEDs and SDs, as predicted by the centripetal pattern of learning in the theoretical model (*Liou et al., 2020*). To address this question, we tested for differences in speed and

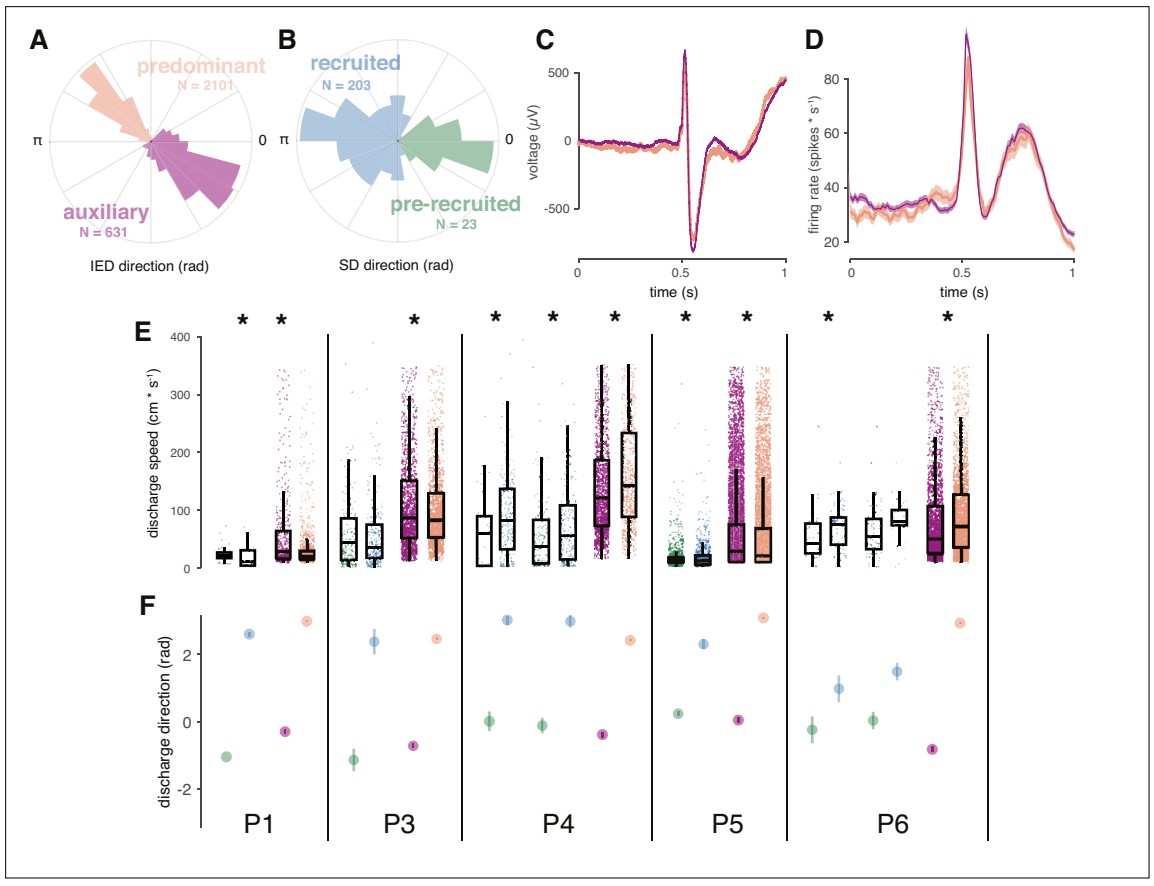

**Figure 5.** Correspondence between spatial features of IED and SD sub-distributions. (**A**) Example IED sub-distributions from one participant. (**B**) Example SD sub-distributions from the same participant as in (**A**). Numbers of IEDs in each sub-distribution are displayed in the color of each sub-distribution. (**C**) Mean IED waveforms from each sub-distribution, color coded as in (**A**). (**D**) Mean firing rates from each IED sub-distribution, color coded as in (**A**). (**E**) Scatter and box plots for IED and SD propagation speeds, separated by sub-distribution across subjects, and color coded as in (**A-B**). Boxes indicate median, quartiles, and whiskers indicate 1.5 times the interquartile range. Speeds greater than 350 cm/s are not shown for display purposes. Asterisks indicate significant differences (p < 0.05). (**F**) Distribution summaries for IED and SD traveling wave directions, separated by sub-distribution across subjects, and color coded as in (**A-B**). For the two participants with more than one seizure (P4, P6), each seizure is shown separately in (**E**) and (**F**).

relative size of SD and IED sub-distributions. Speeds were significantly different between IED sub-distributions within each participant (*Figure 5E*; Mann-Whitney U, all p < 10⁻⁴). Speeds were also significant between SD sub-distributions in five of the seven seizures (*Figure 5E*; Mann-Whitney U, p < 0.04 in five seizures; p > 0.62 in two seizures). Finally, the proportion of IED directions in each sub-distribution predicted the direction of each SD sub-distribution in four of the five participants (two sample proportion tests, $\chi^2$ > 347.6, p < 10⁻⁶). More pre-recruitment discharges occurred in a fifth patient with nearly equivalent proportions of IEDs across sub-distributions. Importantly, the directions of significant differences in these spatial features corresponded across IED and SD sub-distributions. These results show that when IED and SD distributions were bimodal, their spatial features were similar, underscoring the extent to which IEDs mimic spatiotemporal features of ictal self-organization.

## Discussion

Our results, using microelectrode array recordings in patients with epilepsy, show that IEDs are traveling waves that echo – and are predictive of – the propagation patterns of SDs, along a parallel axis to the direction of expansion of the seizure core. It is likely that the frequent barrages of coordinated activity seen during seizures—SDs—potentiate propagation pathways through neocortex that are revisited during IEDs. This conclusion is corroborated by the predictions of our theoretical work,

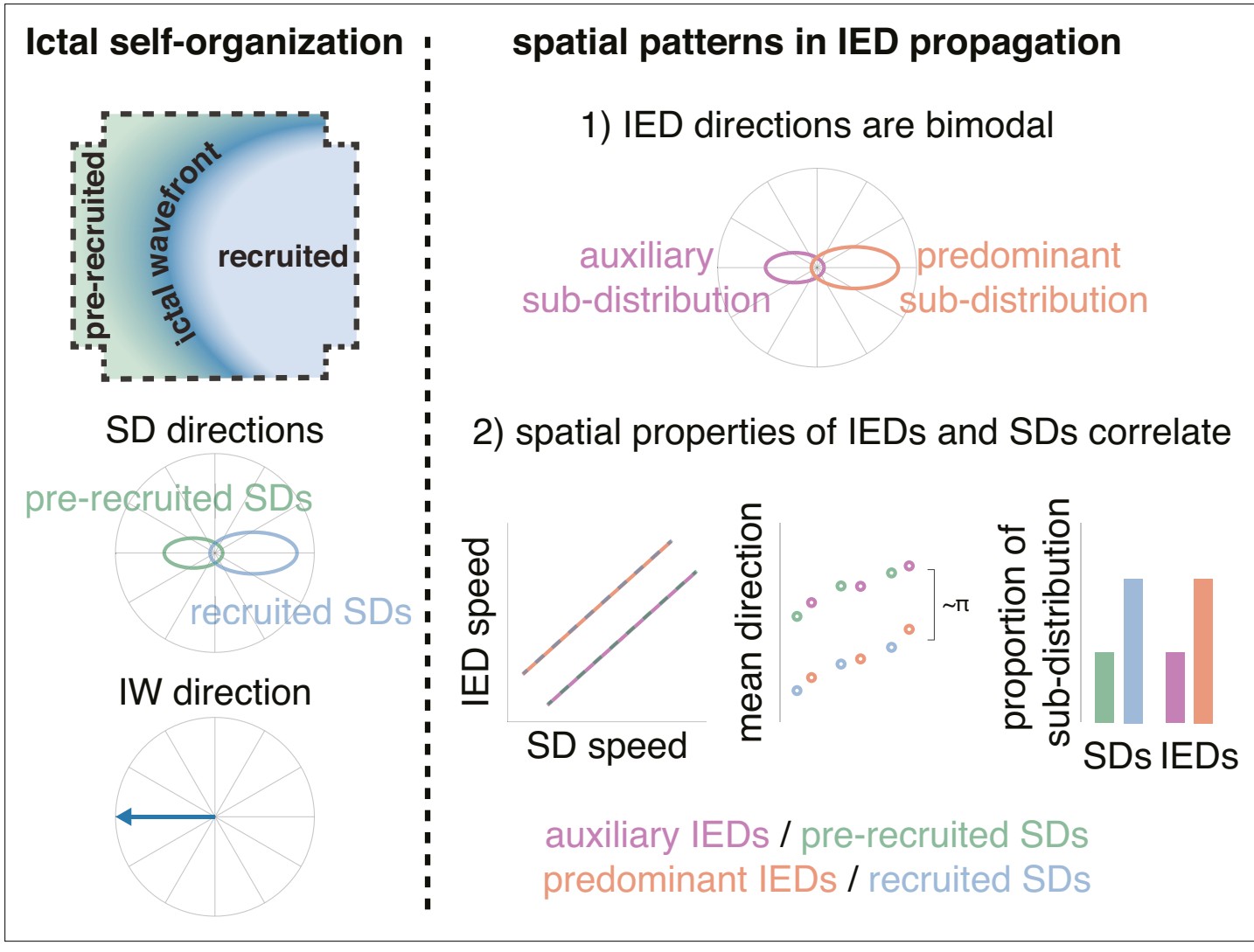

**Figure 6.** Schematic of how IEDs relate to ictal self-organization. A schematic of spatial features of ictal self-organization is shown on the left. Schematics illustrating the correspondence between spatial features (speed, directions, and proportion) of IEDs and SDs is shown on the right.

where a computational model incorporating spike-timing dependent plasticity and realistic connectivity between inhibitory and excitatory cells self-organized to produce IWs, SDs, and IEDs that echo through the pathways potentiated by the strong, repeated barrages of SD activity, antipodal to the IW direction (*Figure 6*; *Liou et al., 2020*; *Nguyen et al., 2020*). Therefore, these results suggest that IEDs originate from tissue immediately eccentric to the seizure core and propagate inwardly across the seizure onset zone, indicating their potential use for localizing the seizure source (*Liou et al., 2020*; *Liou et al., 2019*).

The empirical results reported here extend our understanding of the geometric properties of epileptic tissue, beyond the model predictions, in showing that IEDs travel largely bidirectionally on a linear axis. The bidirectional propagation of IEDs is similar to the bidirectional traveling waves we have previously observed during seizures (*Liou et al., 2017*; *Smith et al., 2016*). The bidirectional pattern of traveling waves during seizures emanated from a slowly expanding, motile source of ictal activity—the IW—passing through a fixed recording site (*Smith et al., 2016*). The data presented here show that IEDs travel in similarly oriented, bimodal distributions, even in the absence of an IW or ictal self-organization.

The IED directions we report are not perfectly antipodal to the IW. These differences in mean angles may be due in part to error when fitting a plane to the IW, which discounts the propagating wave's curvature, or biological variability, such as changes in brain state and arousal. Furthermore, much of the deviation in the summary statistics is driven by one participant (P3). Rigorous statistical analysis of cluster bimodality, the resulting mean angles from that analysis, and KLD between IEDs and SDs all support the major result of the paper, that IEDs and SDs travel bidirectionally along a similar path. The dominant direction of travel is largely in the opposite direction of seizure expansion.

The theoretical model predicted that the directional preferences of IEDs were learned from SDs via spike-timing dependent plasticity (*Bi and Poo, 1998*). While we cannot address the specific learning mechanism with this dataset, in participants whose UEAs were recruited into the seizure core from adjacent cortical tissue, we showed correspondences between several spatial properties of IEDs and SDs. The speeds, directions, and relative sizes of IED sub-distributions echoed those of SDs. Moreover, predominant IED and SD directions opposed the directions of the ictal wavefronts in 'recruited' UEAs. These relationships were unable to be determined from UEAs that were not recruited into the seizure core ('penumbral' recordings). Together, these results suggest that spatiotemporal biases exist in epileptic tissue. Whether spatiotemporal biases in IEDs arise from learning during SDs or vice versa remains to be determined. While the theoretical model indicates that several seizures must occur before IEDs begin to form, electrographic discharges, similar to IEDs, often appear before seizures in animal models of epilepsy (*Staley et al., 2011*).

While we show that the majority of IEDs are traveling waves whose directions overlap with those of SDs, it is important to recognize the small spatial scale of the recordings analyzed here. Beyond the small area of brain we recorded from, there is some sampling bias inherent in these recordings, as we sought to place the microelectrode arrays as close to the site of seizure onset as possible, and within the area of resection. Additional, more eccentric populations of IEDs could be propagating from distant areas that are connected to the seizure onset zone, though not necessarily from adjacent tissue on the cortical surface (*Gelinas et al., 2016*). Higher density ECoG that spans a larger cortical territory than the UEA would be useful in gaining more context on where IEDs arise, and how IEDs propagate across the cortical surface. On the other hand, there is currently no evidence that the ictal wavefront can be detected without action potential recordings, though the time of seizure recruitment can be roughly estimated on each ECoG electrode from high-frequency LFP (*Smith et al., 2020*; *Weiss et al., 2013*). Precise IED propagation patterns are also difficult to measure with the relatively low sampling density of ECoG, and minor shifts in the relative orientation of the ECoG grid and microelectrode array could impugn the accuracy of comparing traveling wave directions between the two modalities. Animal studies using calcium indicators capable of imaging neuronal activity across large cortical territories may overcome these limitations (*Liou et al., 2019*; *Wenzel et al., 2017*). Such animal studies are also poised to understand how learning and plasticity contributes to the geometric relationships reported here.

While these MEA recordings inform fundamental geometric relationships between IED propagation and ictal self-organization, several MEAs flanking the seizure onset zone may be required to accurately triangulate the SOZ from traveling wave directions. Additional spatial context may

also inform our understanding of 'penumbral' tissue and its relevant spatial biases (*Smith et al., 2020*). Future work will therefore focus on translating these microelectrode array results to a more clinically relevant spatial scale. For example, using ECoG (*Khodagholy et al., 2015Khodagholy et al., 2015*; *Viventi et al., 2011*) with vector field or convolutional methods (*Muller et al., 2016*), or examining propagation of source-localized IEDs in stereo-EEG along white matter tracts. Such approaches will be useful for linking the micro results reported in this paper to the coarser spatial resolutions and broader coverages encountered with typical intracranial recordings. Recordings with a broader spatial scale could also inform how IEDs propagate in a larger epileptic network, including patients whose seizures have a common onset site: the mesial temporal lobe (MTL). Three of the participants in this study (P1, P2, and P3) were classified as having MTL seizure onsets from their clinical reports, yet still exhibited similar IED and SD traveling wave dynamics. Despite MTL onsets, these patients' seizures spread to the lateral temporal cortex, and eventually to the microelectrode array site in all three cases. We would speculate that this lateral seizure spread in neocortex could establish similar spatiotemporal biases to those in the other patients, yet a more widespread recording technology with higher resolution than ECoG may be required to full reveal the dynamics of MTL to neocortex seizure spread. It is noteworthy that two of these patients had larger differences between median IED and SD direction distributions than the others. Integrating these multi-scale geometrical understandings of how IEDs relate to the seizure onset zone could then provide an additional piece of information to inform diagnosis and treatment of medically refractory epilepsy, and potentially enable localizing the seizure source without having to directly observe seizures.

# Materials and methods

**Key resources table**

| Reagent type (species) or resource | Designation | Source or reference | Identifiers | Additional information |
|---|---|---|---|---|
| Software, algorithm | MATLAB | https://www.mathworks.com/products/matlab.html | SCR:001662 | |
| Software, algorithm | NPMK | https://github.com/BlackrockNeurotech/NPMK (*Torab, 2014*) | | |
| Software, algorithm | IED analysis code | https://github.com/elliothsmith/IEDs (*Smith, 2022*) | | |
| Software, algorithm | Circular statistics toolbox | https://github.com/circstat/circstat-matlab (*Berens et al., 2019*) | | |
| Software, algorithm | fitmvmdist | https://github.com/chrschy/mvmdist | | |
| Software, algorithm | hrtest | https://github.com/cnuahs/hermans-rasson (*Cloherty, 2020*) | | |

## Participants, ethics statement, and data

The data for this study were acquired from Utah-style microelectrode arrays (UEAs) that were implanted in 10 human patients across two surgical sites who were undergoing neurophysiological monitoring for surgical treatment for medically refractory seizures. Clinical details for all participants are shown in *Appendix 1—table 1*. The Institutional Review Boards at the University of Utah (IRB_00114691) and Columbia University Medical Center (IRB-AAAB6324) approved these studies. All participants provided informed consent prior to surgery for implantation of the clinical electrocorticography (ECoG) electrodes and UEA (10 × 10 electrodes in 4 × 4 mm, penetrating 1 mm). Methodological details of surgical implantation of UEAs into human epilepsy patients area described in detail in *House et al., 2006*. During implantation of ECoG electrodes, UEAs were pneumatically inserted into areas that were most likely to be in the seizure onset zone, and therefore most likely to be resected. Electrophysiological data were pseudodifferentially amplified by 10 and acquired at 30 kilosamples per second using a neural signal processing system (Blackrock Microsystems, Salt Lake City, UT) semi-chronically, that is throughout the duration of the participants' hospital stays. Patients were weaned off their antiepileptic medication during that time, as dictated by their individual clinical courses. Throughout the manuscript, numerical quantities are presented as mean ± standard deviation (μ ±σ).

## IED detection and signal processing

In order to detect IEDs from continuous data recorded on each UEA channel, we developed a simple algorithm for detecting IEDs across a microelectrode array (Appendix 1 - Algorithm S1). For each channel on each UEA, we first resampled the data at 400 samples per second and zero-phase filtered the data between 20 and 40 Hz using a fourth-order Butterworth filter. We then detected any peaks in the absolute amplitude of this signal that were greater than eight times the standard deviation of the remainder of the recording segment (2-hr median duration; *Figure 1—figure supplement 3A*). In order to remove redundant detections, those following any other detection by less than 250ms were discarded. Only detections that occurred within the same 250ms window across at least 10 electrodes on the UEA were retained for further analysis (*Figure 1—figure supplement 3B*). Spectrograms of IEDs were generated via the continuous wavelet transform following the methods used in *Schevon et al., 2012*; *Smith et al., 2020*, and colored with the turbo color map (*Mikhailov, 2021*).

Multiunit action potentials (MUA) were detected on each microelectrode by filtering each channel between 0.3 and 3 kilohertz and detecting peaks in the filtered signal less than –4 times its root mean square. The times of these peaks were retained for further analysis. Example retained detections are shown in *Figure 1—figure supplement 3C,E*.

We employed several post-detection processing steps to ensure the quality of this expansive data set and reject artifacts. First, temporally outlying voltage extrema were removed in order to constrain extrema detection into a temporally focused window (approximately 50 ms duration) around the time of IED detection and to exclude broken microelectrodes or those without IED signal. Next, we excluded any discharges with outlying amplitudes, defined as double the interquartile range of the distribution of IED voltage ranges (*Figure 1—figure supplement 3F,G*).

In order to validate the detection algorithm's ability to detect real IEDs, we calculated the algorithm's positive predictive value (precision) against ratings of experienced clinicians (C.A.S. and J.D.R.) on a dataset of 78 random IED detections, including 9 of those that did not pass our post-detection quality assurance steps described in the previous paragraph. Algorithmic positive predictive value (precision) was defined as the ratio of the number of true positive detections to the number of total positive detections. Cohen's Kappa was also calculated to evaluate inter-rater reliability against chance.

## Traveling wave measurement

In order to measure IED traveling wave speed and direction, we fit a plane to the timings of IED voltage minima, and MUA event times, using ordinary multilinear regression, regularized via the absolute deviation of the signal (*Figure 1—figure supplement 4*). This methodology is described in detail and validated for measuring traveling waves during ictal discharges in *Liou et al., 2017*. Briefly, the regression model for each IED yielded three coefficients, describing the best-fit plane to the timing of IEDs across the UEA in spacetime. Traveling wave direction was determined by the gradient direction of the plane and speed was defined as the inverse of that gradient norm. Each IED was operationally defined as a traveling wave if its model significantly deviated from a plane with zero slope. Statistical significance for this measure was determined by a permutation test in which the model was reevaluated 1,000 times with the microelectrode spatial locations randomly permuted. Differences in IED speed across and within participants were tested using a two-way Kruskal-Wallis Test with a participant factor (10 levels; one for each participant) and a signal factor in which the two levels were the speed measurements derived from LFP and MUA. Only MUA times from 50ms before and after the median time of the LFP negative peak were included in the regression model (*Liou et al., 2017*). Post-hoc pairwise comparisons were carried out using Dunn's test (*Dunn, 1964*). The significance criterion was chosen as 0.05 for all of these tests.

## Directional statistics

Polar histograms were plotted using 18 bins. Circular normal distributions were fit using the circular statistics toolbox (*Berens, 2009*). These distributions defined by two parameters, μ and κ, which describe the central angle and concentration of the distribution, respectively. μ and 1/κ are analogous to the mean and variance parameters that define a standard normal distribution. Directional statistics were carried out using modified functions from the circular statistics toolbox (*Berens, 2009*). These modifications were such that statistical significance was evaluated using permutation

tests from which p-values were derived by comparing the circular test statistic with a distribution of circular test statistics from 1000 permuted datasets. As an example, testing for differences between IED and SD means would involve comparing the test statistic from the true data to a distribution of 1000 test statistics in which the measurement categories were permuted. The significance criterion was chosen to be 0.05. Hypotheses that within-participant IED propagation directions were non-uniform, were tested with Hermans-Rasson tests of circular non-uniformity, again with 1000 permutations (*Landler et al., 2018*).

## Bimodality and sub-distributions

In order to determine whether two, unimodal distributions better fit the ostensibly bimodal IED and SD distributions we observed, we first fit von Mises Mixture (vMM) Models to overall distributions of IED and SD directions using the Matlab function *fitmvmdist* (https://github.com/chrschy/mvmdist; *Schymura, 2016*). Overall IED direction distributions were then clustered into two component vMM distributions using the Matlab function *cluster*. We did not observe distributions that appeared to have more than two modes and therefore set an upper limit on the number of hypothesized clusters, $h$, at two. These vMMs yielded three parameters for each sub-distribution, $h \in N$, such that $h \leq 2$: the sub-distribution means, $\mu_h$, concentration parameters, $\kappa_h$, and probability densities, $\theta_h$.

Rather than assuming these vMM models better fit overall IED and SD distributions, we assessed whether the overall distribution or each vMM sub-distribution better fit the distributions defined by $\mu_h$ and $\kappa_h$. The permutation-based Kuiper tests used to assess goodness-of-fit were carried out as follows. We first estimated $\mu_h$, $\kappa_h$, and $\theta_h$ for both the overall and vMM sub-distributions. We then carried out permutation-based Kuiper tests, to compare empirical distributions of 60 randomly sampled IED directions to theoretical circular normal distributions derived from $\mu_h$ and $\kappa_h$ from both the original and vMM sub-distributions. We repeated this procedure 1,000 times in order to create a permutation distribution. In this way, we were able to measure the extent to which randomly sampled IED angles deviated from theoretical circular distributions defined by the overall and vMM parameters. This is akin to cross-validating vM parameters and choosing $h$ corresponding to the highest log-likelihood, yet in a model-free way. Comparing Kuiper test statistics to a permutation distribution, rather than zero (the null hypothesis of a uniform distribution), makes the tests more conservative and allows us to determine whether the distribution cannot be determined to be non-normal (*Louter and Koerts, 1970*). We then defined our circular bimodality index as the minimum difference between the Kuiper test statistic for the overall distribution and each vMM sub-distribution. Positive bimodality indices thus indicated that overall traveling wave distributions were better modeled as two vMM sub-distributions, and negative bimodality indices indicated that overall traveling distributions were better modeled as a single von Mises distribution. The Matlab functions for implementing these classifications are highlighted in the online code repository.

## Seizure characterization

In order to study the propagation patterns of IEDs relative to seizures, it was necessary to quantify spatial features of SDs and seizure expansion for each recorded seizure. These measures have also been described in previous publications (*Smith et al., 2020; Schevon et al., 2012; Smith et al., 2020; Smith et al., 2016*). The ictal wavefront (IW) is the slowly expanding edge of the seizure representing the spatial signature of failure of feedforward inhibition, and therefore defines recruitment of the tissue surrounding an electrode into the ictal core (*Schevon et al., 2012*). This biomarker of seizure recruitment and expansion has thus far only been detected with recordings of multiunit firing rates (*Smith et al., 2016*), although can also be detected on single microelectrodes by observing widening of action potential waveforms (*Smith et al., 2020; Merricks et al., 2015*). We followed methods from our previous manuscripts to generate multiunit firing rates, that is filtering the broadband data between 300 and 3000 Hz and detecting any peaks larger than the median absolute value of the signal divided by 0.6745 (*Quiroga et al., 2004*).

To find seizures, we examined time periods in the microelectrode array recordings that corresponded to the seizure times reported in the clinical monitoring reports and confirmed that the microelectrode array LFP exhibited the characteristic high amplitude, rhythmic discharging associated with seizures. We then looked for tonic multiunit firing spreading across the array that would suggest the presence of an ictal wavefront, and phase locked multiunit bursting associated with ictal discharges.

We detected the ictal wavefront feature of seizures by smoothing the firing rates on each microelectrode with a 250 ms Gaussian kernel and fitting a multilinear regression model to the peaks of these slow firing rate estimates across the UEA (*Liou et al., 2017*; *Smith et al., 2016*). We detected ictal discharges by detecting peaks in multiunit firing rates, calculated with a 25 ms Gaussian kernel, as we have previously (*Liou et al., 2017*; *Smith et al., 2016*). We quantified the propagation direction and speed of ictal discharges using the same methods used to determine IED traveling wave speeds and directions, as in *Liou et al., 2017*. IED speeds and directions were measured in a manner that was blinded to each microelectrode array's recruitment classification. We defined the presence of ictal phase-locked firing with a Hermans-Rasson test of circular uniformity on MUA action potential times across microelectrodes relative to the phase of the mean LFP recorded across the UEA, similarly to *Schevon et al., 2012*. Using these measures, we operationally defined two patterns of ictal self-organization, based on observed patterns in these fast and slow spatial features of seizures: 'recruited' seizures were operationally defined as those with a significant ictal wavefront multilinear regression model and significant phase-locked multiunit firing (*Figure 3—figure supplement 1A-D*). 'Penumbral' seizures were operationally defined as those seizures in which we were unable to detect an ictal wavefront on the microelectrode array (Fig *Figure 3—figure supplement 1E-H*). Similar classifications of adjacent and non-adjacent recruitment have been reported by other groups (*Martinet et al., 2015*; *Schevon et al., 2012*). For seizures that were associated with secondary generalization, we only included discharges up to the clinically defined point of secondary generalization in order to constrain our study to the dynamics of focal seizure onset and spread.

## Comparing IED and SD sub-distributions

Mean firing rates were estimated by binning MUA event times into one-hundred 10 ms bins across microelectrodes. For distributions that were determined to be bimodal, we used cluster-based permutation tests to test for differences between median IED waveforms and firing rates between IEDs from the two component vMM distributions for each participant. Permutation tests consisted of carrying out t-tests on each sample in the IED waveforms and each bin in the IED firing rates. The t-statistics for each sample or bin were then compared to permutation distributions of t-statistics for each sample or bin in which the sub-distribution labels were shuffled. If any of these p-values exceeded the specified alpha value of 0.05, we considered that sample significant, and included it with adjacent significant samples in a temporal cluster. In order to control for multiple hypothesis testing, we used cluster-based correction over temporal clusters. This procedure involved finding the largest temporal clusters of significant samples or bins in the permutation distributions and comparing those clusters to the largest temporal clusters of significant samples or bins in the real data.

In order to test for different directions of overall distributions and sub-distributions of IEDs and SDs, we used Watson-Williams multi-sample tests for equal means. To test for differences in IED speed between different sub-distributions, we used within-participant Mann-Whitney U tests. In order to test differences in the proportion of IEDs from predominant and auxiliary sub-distributions we used two-sample proportion tests. We employed a significance criterion of 0.05 for all of these tests.

Finally, in order to directly quantify the reduction in uncertainty about the direction of SD travel that can be estimated from observing IED directions, we calculated the Kullback-Leibler Divergence (KLD) between distributions of IED and SD directions (*Kullback and Leibler, 1951*). We estimated the KLD as the expectation of the logarithmic difference between discrete distributions of IED and SD directions on the half-closed interval between 0 and $2\pi$. We interpret the KLD as measuring the information that can be gained about SD directions from observing IED directions. We also tested for whether these KLD values were significantly greater than would be expected by chance by comparing the true KLD values against a distribution of 1000 KLD values generated from directions drawn at random from both IED and SD distributions.

## Acknowledgements

Thanks to the Schevon and Rolston labs.

# Additional information

## Competing interests

Paul House: is affiliated with Neurosurgical Associates, LLC. The author has no financial interests to declare. Guy M McKhann: reports fees from Koh Young, Inc. Sameer Sheth: consulting for Boston Scientific, Abbott, Neuropace, Zimmer Biomet. John D Rolston: reports fees from Medtronic, Inc. The other authors declare that no competing interests exist.

## Funding

| Funder | Grant reference number | Author |
|---|---|---|
| National Institutes of Health | NINDS R21 NS113031 | Elliot H Smith<br>Catherine Schevon<br>John Rolston |
| National Institutes of Health | NINDS K23 NS114178 | John Rolston |
| National Institutes of Health | S10 OD018211 | Catherine Schevon |
| National Institutes of Health | R01 NS084142 | Catherine Schevon |
| American Epilepsy Society | JIA | Elliot H Smith |

The funders had no role in study design, data collection and interpretation, or the decision to submit the work for publication.

## Author contributions

Elliot H Smith, Conceptualization, Data curation, Formal analysis, Funding acquisition, Investigation, Methodology, Project administration, Resources, Software, Supervision, Validation, Visualization, Writing - original draft, Writing – review and editing; Jyun-you Liou, Conceptualization, Investigation, Methodology, Project administration, Writing – review and editing; Edward M Merricks, Investigation, Methodology, Resources, Writing – review and editing; Tyler Davis, Data curation, Investigation, Resources, Writing – review and editing; Kyle Thomson, Conceptualization, Funding acquisition, Investigation, Methodology, Resources; Bradley Greger, Paul House, Data curation, Resources; Ronald G Emerson, Robert Goodman, Guy M McKhann, Resources; Sameer Sheth, Investigation, Resources, Writing – review and editing; Catherine Schevon, Conceptualization, Data curation, Funding acquisition, Investigation, Methodology, Project administration, Resources, Software, Supervision, Validation, Writing – review and editing; John D Rolston, Conceptualization, Funding acquisition, Investigation, Methodology, Project administration, Resources, Supervision, Writing – review and editing

## Author ORCIDs

Elliot H Smith ![ORCID] http://orcid.org/0000-0003-4323-4643
Jyun-you Liou ![ORCID] http://orcid.org/0000-0003-4851-3676
Edward M Merricks ![ORCID] http://orcid.org/0000-0001-8172-3152
Catherine Schevon ![ORCID] http://orcid.org/0000-0002-4485-7933
John D Rolston ![ORCID] http://orcid.org/0000-0002-8843-5468

## Ethics

Human subjects: The Institutional Review Boards at the University of Utah (IRB_00114691) and Columbia University Medical Center (IRB- AAAB6324) approved these studies. All participants provided informed consent prior to surgery for implantation of the clinical and research electrodes.

## Decision letter and Author response

Decision letter https://doi.org/10.7554/eLife.73541.sa1
Author response https://doi.org/10.7554/eLife.73541.sa2

## Additional files

### Supplementary files
• Transparent reporting form

### Data availability

Raw data is available upon establishment of a data use agreement with Columbia University Medical Center as required by their Institutional Review Board (IRB). Data from human subjects was analyzed, from which the dates of implants can potentially be reconstructed. This is especially true for a study like this one, in which chronic recordings were carried out for the full duration of the patients' hospital stays. Sharing these data widely could therefore expose private health information of participants, which is why a data use agreement is required by the IRB. Interested Researchers should contact Dr. Schevon to get the data use agreement process started with the Columbia University Medical Center IRB. Analysis code is upload to GitHub: https://github.com/elliothsmith/IEDs, (copy archived at swh:1:rev:218f6e60e8f0b32d4e16a88298475618d5c09589). We have included preprocessed data files for all IEDs, hosted online at OSF: https://osf.io/zhk24/. Data files include LFP, MUA event times, and traveling wave model coefficients for all detected IEDs.

The following dataset was generated:

| Author(s) | Year | Dataset title | Dataset URL | Database and Identifier |
|---|---|---|---|---|
| Smith EH, Liou JY, Merricks EM, Davis TS, Thomson K, Greger B, House PA, Emerson RG, Goodman RR, McKhann II GM, Sheth SA, Schevon CA, Rolston JD | 2021 | Human interictal epileptiform discharges are bidirectional traveling waves echoing ictal discharges | https://osf.io/zhk24/ | Open Science Framework, osf.io/zhk24/ |

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

## Appendix 1

**Appendix 1 - Algorithm 1. Algorithm for detecting IEDs from microelectrode recordings.**

Input: A matrix of microelectrode voltage recordings, V(c, t), where measured voltage is a function of time, t, and which microelectrode array channel it was recorded on, c.

Output: A vector of IED times, I(t).

1: for each c, do:

2: filter data between 20 and 40 Hz (non-causal, 4th order Butterworth filter)

3: detect peaks, p, in filtered signal greater than eight times the standard deviation of the beta power in the data.

4: discard peaks that occur within 250ms of a preceding detection

5: for each p, co-occurring within 250ms, across more than 10 microelectrodes.

6: find all local minima of V(c, t) in time, across all c.

7: bin local minima in time, across all c.

8: convolve the resulting histogram with a modified Heaviside function,

$$H(t) : \begin{cases} 0, 0 \geq n < 0.4 \\ 10t, 0.4 \geq n < 0.5 \\ -t, 0.5 \geq n < 1 \end{cases}$$

9: find absolute minima of V(c, t) within the bin containing the most local minima.

**Appendix 1—table 1.** Clinical details for research participants.

| Participant | Age | Sex | Epileptogenic zone | UEA implant site | Pathology | Outcome |
|---|---|---|---|---|---|---|
| 1 | 19 | female | right posteriror lateral temporal | Right posterior temporal, 1 cm inferior to angular gyrus | Non-specific | Engel 1 a at >2 years |
| 2 | 32 | female | left inferior temporal lobe | inferior temporal gyrus, 2.5 cm from temporal pole | one neuronal loss; lateral temporal nonspecific | Engel 1 a at 55 months |
| 3 | 32 | male | Right mesial temporal lobe | right inferior frontal gyrus | Normal hippocampus | Engel1a at 2.5 years |
| 4 | 28 | male | left dorsal posterior prefrontal cortex | left posterior middle frontal gyrus | Mild reactive astrogliosis, patchy microgliosis, Chaslin's marginal sclerosis | Engel 3 a at 32 months |
| 5 | 26 | male | left middle subtemporal | left posterior inferior temporal gyrus | Diffusely infiltrating low grade glioma, IDH-1 negative | Engel 4 a at 2 years, 5 months |
| 6 | 25 | male | Left mesial temporal lobe | Left middle temporal gyrus, 3 cm posterior to temporal pole | Non-specific | Engel 1 a at 7 months |
| 7 | 30 | male | right middle inferior temporal gyrus | right middle temporal gyrus | Mild astrocytosis | Engel 1 a at 12 months |
| 8 | 30 | male | right lateral and mesial temporal lobe; nonlesional | Right middle temporal gyrus, 4 cm posterior to the temporal pole | Mesial temporal sclerosis | Engel 2 at 22 months |
| 9 | 30 | male | left supplementary motor area | left supplementary motor area | N/A (multiple subpial transections performed) | Engel three at >2 years |
| 10 | 39 | male | left frontal operculum | left lateral frontal, 2 cm superior to Broca's Area | Nonspecific | Engel 1 a at >2 years |

