## [Editor Report]

This manuscript describes the propagation patterns of electrical activity in the brains of patients with drug-resistant epilepsy. Specifically, the authors demonstrate that interictal spikes, commonly observed electrical events in epileptic patients, propagate in a similar manner to seizures, which are relatively uncommon and more difficult to capture. This suggests that interictal spikes could be used in surgical planning, improving the localization and treatment of epileptic networks.

---

## [Decision Letter]

**Decision letter after peer review:**

Thank you for submitting your article "Human interictal epileptiform discharges are bidirectional traveling waves echoing ictal discharges" for consideration by *eLife*. Your article has been reviewed by 3 peer reviewers, and the evaluation has been overseen by a Reviewing Editor and John Huguenard as the Senior Editor. The following individual involved in review of your submission has agreed to reveal their identity: Brian Litt (Reviewer #1).

If possible, please address all of the reviewer concerns discussed below. A summary of the essential revisions are listed here.

Essential revisions:

1)Please describe how you validated automated IED detections. If possible, please quantify the positive predictive value for this method for a random subset of IEDs.

2) The angle between the SDs (ictal wavefronts) and IEDs was not identical (exactly opposed). In addition, the SD distributions were not exactly antipodal. Please address your interpretation of the data and whether SDs and IEDs follow the same path and what might account for the sizable angle between their observed paths.

3)Please discuss the sizable proportion of IEDs (~35%) that were not classified as traveling waves.

4)Please describe how seizures were defined.

5)Please elaborate on the implications of antipodal IED distributions.

6)Please provide more detail about how you quantified differences in the IED waveforms for the cluster-based permutation test.

7) Please comment on whether the "mixture" von Mises distribution has more adjustable parameters than the single von Mises distribution model. If so, does this bias your quality of fit towards the bimodal distribution for IEDs?

8) Please further clarify the terms used for the different discharges examined and

discussed in the manuscript.

9) Please compare IED propagation measured from the ECoG array to the IED propagation vector produced from the UEA.

10) Please address whether the resulting surgical resections in recorded patients agreed with the axis predicted by the IED propagation measured by the UEA. If possible, please discuss whether the success of the surgery correlated with the UEA-predicted focus.

11) Please comment on the relationship between the neocortical propagation measured by the UEA on the cortical surface and the ability to accurately localize deep foci such as in MTLE. How well does the bidirectional IED hypothesis apply to mesial temporal lobe seizures where the ictal onset is far from the neocortex?

12) Please include whether patients were weaned from their AEDs during these recordings.

13) Please discuss how the IED distributions for subjects with multiple seizures were calculated.

*Reviewer #1 (Recommendations for the authors):*

1)How did you validate automated IED detections? Could you report on positive predictive values for a random subset of detections?

2)You write in the abstract and discussion that IEDs traverse the same path as ictal discharges, but the angle between the SDs and IEDs was 24 degrees, and the IEDs weren't exactly opposite the ictal wavefront (150 degrees). Do you think the SDs and IEDs really follow the same path, and if so, what accounts for the sizable angle between their observed paths?

*Reviewer #2 (Recommendations for the authors):*

1. The terminology used to describe the various types of epileptiform activity described in this paper should be laid out in a clear way from the beginning. Although many of the features are defined in prior work, they are critical to understanding the results. Specifically, the introduction of SDs focuses on their propagation following the IW, but "pre-recruitment" SDs are a prominently analyzed group in Figure 5. It would be helpful to include a schematic figure like that at the top left of 6A earlier in the paper. It may also be useful to define two types of SDs (anterograde and retrograde?), and include them on the schematic figure. Perhaps "recruited" and "penumbral" could be illustrated here as well (or in a separate subpanel).

2. The prominent clinical relevance of the analysis presented in this manuscript is that IEDs could be used to locate the seizure focus in pre-surgical planning. As briefly indicated in the discussion, the UEA is likely too small to localize a seizure focus – it produces one vector, which at best could define the line along which the focus falls. To identify the seizure focus, greater spatial sampling is needed. The authors point out that the simultaneously recorded ECOG is insufficient to measure IW propagation as it cannot detect MUAs and its poor spatial sampling make estimation of IED propagation low-fidelity. Nevertheless, since LFP was used to record IED propagation in the UEA, it would be useful to analyze IED propagation from the ECoG array to compare the vector produced, particularly by neighboring electrodes.

3. If the ECoG analysis fails to produce a vector that match the UEA vectors, it may be necessary to implant two UEAs. The intersection of the IED propagation vectors would hypothetically localize the seizure focus (in 2D). Obviously this experiment is outside the scope of this study, but worth mentioning in the discussion.

4. Did the resulting surgical resections in recorded patients align with the axis predicted by the IED propagation at the UEA? Did the success of the surgery relate to the removal (or not) of the UEA-predicted focus?

5. Synaptic connectivity is probably important determinant of whether vector points to focus. Furthermore, UEA recordings are performed on the cortical surface. In the most straightforward application of the proposed strategy, seizures would start on cortical surface and propagate locally. However, many seizures begin in deep structures. Some discussion of the relationship between neocortical propagation and localization of a deep focus (e.g. MTLE) is warranted.

6. Were patients weaned from their AEDs during these recordings. One important advantage of analyzing IEDs vs seizures for surgical planning is that they are very common. Thus, it might be possible to perform the mapping in patients without having them in withdrawal from their AEDs, which might improve predictive value.

7. In Figure 4A-F, for the subjects with multiple seizures, the IED distribution appears to be different for each seizure. What IED distributions are plotted – only those that had occurred up to the time of that seizure? Maybe I missed it, but I couldn't find this info in the results or figure caption…

*Reviewer #3 (Recommendations for the authors):*

The Utah array is an excellent tool to study spatiotemporal dynamics in and around a seizure core. The strength of such a technique is unfortunately also the weakness. Since it is implanted on the cortex, the modeling of IED and seizure discharges has a potential for bias due to the location of the implant. The model can be strengthened by comparing the MUA and LFP of the model with their single-unit data of IEDs from hybrid electrodes of mesial temporal lobe implants. The authors do mention the limitation of their study, but it would help if they discuss this some more, especially since only two patients in Table S1 had mesial temporal onset and the rest were lateral or neocortical onsets. They should not generalize that this model is applicable to all seizure types.

[Editors' note: further revisions were suggested prior to acceptance, as described below.]

Thank you for resubmitting your work entitled "Human interictal epileptiform discharges are bidirectional traveling waves echoing ictal discharges" for further consideration by *eLife*. Your revised article has been evaluated by John Huguenard as the Senior Editor, and a Reviewing Editor.

The manuscript has been improved but there are some remaining issues that need to be addressed, as outlined below:

The majority of reviewer concerns have been adequately addressed. However, there remain a few points that require clarification or additional submission materials. The Reviewing Editor has drafted this concise list to facilitate your revision.

Essential revisions:

1) Multiple reviewers raised the question regarding "the sizable angle between… SDs and IEDs" and the fact that the "IEDs weren't exactly opposite the ictal wavefront". You state that "the angle discrepancies to which the reviewer refers are statistically insignificant and can be explained by expected biological variability". On the other hand you state that based on Smith et al. 2016, you don't expect the IED to be strictly antipodal to IW. Finally, while the KLD analysis is useful it does not seem to fully address the reviewers' questions regarding what appears to be a (statistically) significant deviation (almost 2 standard deviations) of the IED/IW angle from 180 degrees. A single paragraph in the discussion summarizing and clarifying the various points of why SD, IEDs, and IWs may not be strictly aligned or antipodal would improve the manuscript.

2) Regarding Reviewer 2, major point #7 "In Figure 4A-F, for the subjects with multiple seizures, the IED distribution appears to be different for each seizure. What IED distributions are plotted – only those that had occurred up to the time of that seizure?" The reviewer is asking why the shape of the distribution, and not just the rotation, of IED distribution varies for different seizures in the same patient. Are you analyzing/including a different subset of IEDs for each seizure? If so, why? If not, why does the distribution change?

3) Regarding Reviewer 2, major point #2, we strongly suggest including the figure showing the "qualitative information about the location of grids, arrays, and IED vectors". We realize that there are a number of caveats and limitations in the interpretation of this qualitative figure that require further explanation, but including the figure would greatly enhance the clarity of the manuscript

4) Please refer to https://submit.elifesciences.org/html/*eLife*_author_instructions.html#revised for instructions on how to submit source data for each figure. There do not seem to be source data included for the majority of figure plots. Also the data repository on OSF would greatly benefit from a description of the different data file types, the different variable types within the structures, and the sampling rate for the IED traces.

---

## [Author Response]

Reviewer #1 (Recommendations for the authors):1) How did you validate automated IED detections? Could you report on positive predictive values for a random subset of detections?

We have validated the algorithm against the gold standard (board-certified neurologist) and calculated the PPV for a subset of 78 IEDs across patients. Two clinicians who were adept at identifying IEDs evaluated groups of 10 microelectrode array channels along with a raster of firing across channels for 10 s segments around each IED (CAS and JDR). We calculated the positive predictive value against the ratings of the board-certified neurologist (CAS). The PPV for the neurosurgeon was 0.83, whereas the PPV for the algorithm was 0.89. Calculating the PPVs for both clinicians in reference to the algorithm yielded 0.89 for both the neurosurgeon and neurologist. We also calculated Cohen’s Kappa to measure interrater reliability against chance performance, and report a value of 44.9%, which is similar to kappa values (48.6%) reported in a large multicenter study of IED interrater reliability (Jing et al., 2020). Cohen’s Kappa between the algorithm and Neurologist was 61.1%. These results are reported in the Results section on page 4, lines 16-20, and the method of validation is described in the methods on page 13, starting on line 8.

2) You write in the abstract and discussion that IEDs traverse the same path as ictal discharges, but the angle between the SDs and IEDs was 24 degrees, and the IEDs weren't exactly opposite the ictal wavefront (150 degrees). Do you think the SDs and IEDs really follow the same path, and if so, what accounts for the sizable angle between their observed paths?

In testing the hypotheses that arose from the model predictions, we assumed that a tolerance of approximately π/4 would be appropriate to report that IEDs and SDs were similar directions, as we showed there is not a directly antipodal relationship between pre and post -recruitment SDs (Smith et al., 2016). However, the reviewer is correct to point out that such a heuristic is insufficient. We therefore carried out statistical tests to determine the extent to which the similarity of these directions might arise by chance. This test relied on comparing the true KLD between IED and SD distributions to a permutation distribution comprised of randomly sampled directions from both distributions. This test is described in the methods on page 16, starting on line 15. All of these tests were highly significant (all p < 0.01), indicating that while the IED/SD distributional overlap is not perfect, it is significantly unlikely to occur by chance in each patient. We have reported these results on page 8, on lines 9-12.

It's also important to note that the mean angle differences the reviewer points to represent those across participants, where two patients had larger differences, but the majority had mean angle differences near zero, as well as mean IED angle differences from the IW direction near 180 (Figures 4 G-H). This many similar angle differences, across all but 2 patients (in which the angle is only slightly larger) is highly unlikely to occur by chance. The two patients with larger angles also were classified as MTL onset, which we have mentioned in the final paragraph of the discussion (Page 11, sentence starting on line 45).

Reviewer #2 (Recommendations for the authors):1. The terminology used to describe the various types of epileptiform activity described in this paper should be laid out in a clear way from the beginning. Although many of the features are defined in prior work, they are critical to understanding the results. Specifically, the introduction of SDs focuses on their propagation following the IW, but "pre-recruitment" SDs are a prominently analyzed group in Figure 5. It would be helpful to include a schematic figure like that at the top left of 6A earlier in the paper. It may also be useful to define two types of SDs (anterograde and retrograde?), and include them on the schematic figure. Perhaps "recruited" and "penumbral" could be illustrated here as well (or in a separate subpanel).

We have edited the second paragraph of the introduction to make this clearer, and have added a supplementary figure (Figure 1—figure supplement 1) similar to that in (Smith et al., 2020) in order to clarify the seizure domains of interest. In that paragraph, we have added quotations to bring the reader’s attention to the pre-recruitment and post-recruitment discharge terms that correspond to the reviewer’s idea of anterograde and retrograde discharges. We showed previously that discharges that precede the wavefront largely travel in the same direction as the IW, discharges that follow the IW largely travel antipodally (Smith et al., 2016).

2. The prominent clinical relevance of the analysis presented in this manuscript is that IEDs could be used to locate the seizure focus in pre-surgical planning. As briefly indicated in the discussion, the UEA is likely too small to localize a seizure focus – it produces one vector, which at best could define the line along which the focus falls. To identify the seizure focus, greater spatial sampling is needed. The authors point out that the simultaneously recorded ECOG is insufficient to measure IW propagation as it cannot detect MUAs and its poor spatial sampling make estimation of IED propagation low-fidelity. Nevertheless, since LFP was used to record IED propagation in the UEA, it would be useful to analyze IED propagation from the ECoG array to compare the vector produced, particularly by neighboring electrodes.

While we agree with multiple elements of this critique, there are numerous reasons why we believe that a full analysis of IED propagation patterns in ECoG is beyond the scope of this study. A primary issue is that the brain sampled by a planar Utah array can be readily mapped to 2-dimensional space, as in this study. However, the brain beneath a grid or sampled by SEEG is not flat—the sampling is no longer along a rigid 2-dimensional plane, given the brain’s gyri and sulci, so Euclidean distances are not an appropriate metric for relating recording locations to brain activity. We are currently working to understand mesoscale patterns in IED propagation on sEEG and ECoG using methods that are more appropriate for: (1) the larger spatial scale, (2) complex 3D patterns of IED travel involved with sEEG/ECoG (e.g. travel across Euclidean, geodesic, or axonal lengths), and (3) more complicated IED waveforms recorded on ECoG and sEEG. There are further methodological uncertainties inherent in spatiotemporal alignment between ECoG and UEA IED propagation vectors that made us uncomfortable with making claims about the correspondence between ECoG and MEA IED propagation. Temporally, alignment was not carried out similarly across all 10 participants (given the different recording locations and approximately a decade between some recordings). Spatially, it is nearly impossible to *precisely* align the MEA and ECoG from intraoperative photos and CT/MRI co-registration due to ‘brain drop’ or potential spatial translation/rotation of ECoG grids after surgery. Physical memory in the UEA wire bundle could add systematic bias to angle estimation from CT/MR co-registration.

Nevertheless, we have included supplementary qualitative information about the location of grids, arrays, and IED vectors for reviewers to judge. Figure 4—figure supplement 2 shows electrode localization information for each participant, a schematic of the implant around the UEA, and rotated IED distributions to match each UEA orientation. While this is all qualitative information, as we cannot precisely align IED distributions based on intraoperative photos, the axis of IED travel bisects the SOZ in most cases.

Furthermore, we have done some preliminary analysis of peri-ictal IEDs on ECoG for one of the penumbral participants, and can share one patient’s results in Author response image 1.

**Author response image 1. sa2fig1:** For one participant whose UEA was not recruited into the seizure examined gradients in the beta phase across the ECoG grid space during the IEDs preceding seizures. We evaluated the divergence of these gradients in order to determine sources and sinks in patterns of IED traveling waves, hypothesizing that the location of the ictal core will be a sink.

This figure shows that there is a ‘sink’ IED propagation pattern across an ECoG grid that co-locates with the SOZ. In upcoming work, we aim to translate these methodologies to sEEG, which is more commonly used in contemporary clinical practice.

Finally, while we recognize the importance of the clinical implications of this work, we would categorize this study as basic science with great relevance for a particular clinical population. We believe it is essential to translate the understandings in this manuscript to a clinically useful scale, and are taking steps to do just that to be reported in future publications. We have added more text to the discussion to clarify the directions that could be taken in order to improve patients’ lives.

3. If the ECoG analysis fails to produce a vector that match the UEA vectors, it may be necessary to implant two UEAs. The intersection of the IED propagation vectors would hypothetically localize the seizure focus (in 2D). Obviously this experiment is outside the scope of this study, but worth mentioning in the discussion.

This is an astute observation by the reviewer. We would refer the reviewer to the schematics on the previous page showing the axis of IED travel bisects the SOZ. We have also added text to the last paragraph of the discussion mentioning the considerations involved with estimating seizure spread from IEDs recorded on MEAs on page 11, line 28.

4. Did the resulting surgical resections in recorded patients align with the axis predicted by the IED propagation at the UEA? Did the success of the surgery relate to the removal (or not) of the UEA-predicted focus?

Appendix 1 – Table 1 shows all the clinical details for each participant. While information about outcomes is in the table there was insufficient statistical power to carry out across subjects analyses (power = 0.14 for a McNemar test). Three patients had outcomes that were Engle 3/4. Arrays in each of those patients were near the SOZ and angles of IED travel bisected the SOZ in each case, however it must be reiterated that it is difficult to infer much from these maps due to uncertainty in spatial alignment of arrays and classification of the SOZ.

5. Synaptic connectivity is probably important determinant of whether vector points to focus. Furthermore, UEA recordings are performed on the cortical surface. In the most straightforward application of the proposed strategy, seizures would start on cortical surface and propagate locally. However, many seizures begin in deep structures. Some discussion of the relationship between neocortical propagation and localization of a deep focus (e.g. MTLE) is warranted.

The third reviewer also suggested we discuss the implications of mesial temporal lobe onsets, so we have added text to the discussion on page 11, starting on line 36 discussing this point directly.

6. Were patients weaned from their AEDs during these recordings. One important advantage of analyzing IEDs vs seizures for surgical planning is that they are very common. Thus, it might be possible to perform the mapping in patients without having them in withdrawal from their AEDs, which might improve predictive value.

Patients were weaned from their AEDs during this study, as dictated by each patient’s clinical course. The IED rate reported in this study may therefore reflect a slightly higher rate than would be expected when patients are fully on AEDs, however the analysis period also includes earlier segments of data that are within the period of IED weaning. We have added a relevant sentence to the methods on page 12, line 25.

7. In Figure 4A-F, for the subjects with multiple seizures, the IED distribution appears to be different for each seizure. What IED distributions are plotted – only those that had occurred up to the time of that seizure? Maybe I missed it, but I couldn't find this info in the results or figure caption…

We thank the reviewer for the opportunity to clarify this point. The figure caption states that each distribution is plotted relative to the direction of the ictal wavefront (IW), which is why all of the IWs point to zero. This is to allow for comparison between IEDs and SDs for each seizure, relative to each seizure’s predominant geometrically organizing feature: the IW. Raw IED and SD distributions, and raw IW directions are plotted in Figure 4—figure supplement 1. We have edited the figure caption for 4A-F in order to clarify both points.

Reviewer #3 (Recommendations for the authors):The Utah array is an excellent tool to study spatiotemporal dynamics in and around a seizure core. The strength of such a technique is unfortunately also the weakness. Since it is implanted on the cortex, the modeling of IED and seizure discharges has a potential for bias due to the location of the implant. The model can be strengthened by comparing the MUA and LFP of the model with their single-unit data of IEDs from hybrid electrodes of mesial temporal lobe implants. The authors do mention the limitation of their study, but it would help if they discuss this some more, especially since only two patients in Table S1 had mesial temporal onset and the rest were lateral or neocortical onsets. They should not generalize that this model is applicable to all seizure types.

We thank the reviewer for the recommendations and have expanded discussion of these points in the final discussion paragraph. Another reviewer also asked us to discuss non-neocortical onset seizures, so we have discussed this on page 11, starting on line 36.

References:

Jing J, Herlopian A, Karakis I, Ng M, Halford JJ, Lam A, Maus D, Chan F, Dolatshahi M, Muniz CF, Chu C, Sacca V, Pathmanathan J, Ge W, Sun H, Dauwels J, Cole AJ, Hoch DB, Cash SS, Westover MB. 2020. Interrater Reliability of Experts in Identifying Interictal Epileptiform Discharges in Electroencephalograms. JAMA Neurology 77:49–57. doi:10.1001/jamaneurol.2019.3531

Liou J, Smith EH, Bateman LM, Bruce SL, McKhann GM, Goodman RR, Emerson RG, Schevon CA, Abbott L. 2020. A model for focal seizure onset, propagation, evolution, and progression. eLife 9:e50927.

Liou J-Y, Baird-Daniel E, Zhao M, Daniel A, Schevon CA, Ma H, Schwartz TH. 2019. Burst suppression uncovers rapid widespread alterations in network excitability caused by an acute seizure focus. Brain 142:3045–3058. doi:10.1093/brain/awz246

Schevon CA, Weiss SA, McKhann G, Goodman RR, Yuste R, Emerson RG, Trevelyan AJ. 2012. Evidence of an inhibitory restraint of seizure activity in humans. Nat Commun 3:1060. doi:10.1038/ncomms2056

Smith EH, Liou J, Davis TS, Merricks EM, Kellis SS, Weiss SA, Greger B, House PA, McKhann II GM, Goodman RR. 2016. The ictal wavefront is the spatiotemporal source of discharges during spontaneous human seizures. Nature communications 7:1–12.

Smith EH, Merricks EM, Liou J-Y, Casadei C, Melloni L, Thesen T, Friedman DJ, Doyle WK, Emerson RG, Goodman RR, McKhann GM, Sheth SA, Rolston JD, Schevon CA. 2020. Dual mechanisms of ictal high frequency oscillations in human rhythmic onset seizures. Scientific Reports 10:19166. doi:10.1038/s41598-020-76138-7

[Editors' note: further revisions were suggested prior to acceptance, as described below.]

Essential revisions:1) Multiple reviewers raised the question regarding "the sizable angle between… SDs and IEDs" and the fact that the "IEDs weren't exactly opposite the ictal wavefront". You state that "the angle discrepancies to which the reviewer refers are statistically insignificant and can be explained by expected biological variability". On the other hand you state that based on Smith et al. 2016, you don't expect the IED to be strictly antipodal to IW. Finally, while the KLD analysis is useful it does not seem to fully address the reviewers' questions regarding what appears to be a (statistically) significant deviation (almost 2 standard deviations) of the IED/IW angle from 180 degrees. A single paragraph in the discussion summarizing and clarifying the various points of why SD, IEDs, and IWs may not be strictly aligned or antipodal would improve the manuscript.

We thank the reviewers for engaging with this issue. We have added an additional paragraph to the discussion to address this issue directly. The new paragraph begins on page 10, line 30. and is pasted here:

“The IED directions we report are not perfectly antipodal to the IW. These differences in mean angles may be due in part to error when fitting a plane to the IW, which discounts the propagating wave’s curvature, or biological variability, such as changes in brain state and arousal. […] The dominant direction of travel is largely in the opposite direction of seizure expansion.”

2) Regarding Reviewer 2, major point #7 "In Figure 4A-F, for the subjects with multiple seizures, the IED distribution appears to be different for each seizure. What IED distributions are plotted – only those that had occurred up to the time of that seizure?"The reviewer is asking why the shape of the distribution, and not just the rotation, of IED distribution varies for different seizures in the same patient. Are you analyzing/including a different subset of IEDs for each seizure? If so, why? If not, why does the distribution change?

The same IEDs go into each plot, but the IED directions are rotated by the ictal wavefront for each seizure. The reason they look slightly different is because all the rotated angles are fitting into the same 18 bins as the original data, so the differences in angles per bin very slightly changes the shape of the distribution. For example in figure 4E, the larger IED bin closer to zero in the right panel appears to get split into two smaller bins in the left panel. It may help the reviewer to compare the IED distributions in figure 4 to the raw distributions that are shown in the figure supplement.

3) Regarding Reviewer 2, major point #2, we strongly suggest including the figure showing the "qualitative information about the location of grids, arrays, and IED vectors". We realize that there are a number of caveats and limitations in the interpretation of this qualitative figure that require further explanation, but including the figure would greatly enhance the clarity of the manuscript.

We agree that this figure does aid the manuscript clarity, despite its many uncertainties. We have included an edited version as Figure 4—figure supplement 2.

4) Please refer to https://submit.elifesciences.org/html/eLife_author_instructions.html#revised for instructions on how to submit source data for each figure. There do not seem to be source data included for the majority of figure plots. Also the data repository on OSF would greatly benefit from a description of the different data file types, the different variable types within the structures, and the sampling rate for the IED traces.

We apologize for this omission. We were under the impression that the uploaded code and OSF database were sufficient to reproduce figures. We have remedied this misunderstanding by uploading source data (IED directions) for figures 2 and 4, which are the main data figures in the manuscript to the github repository. We have also added a description of the data types in the OSF repository.